# Temporally Equivariant Contrastive Learning for Disease Progression

## Abstract

Self-supervised contrastive learning methods provide robust representations by ensuring their invariance to different image transformations while simultaneously preventing representational collapse across different training samples. Equivariant contrastive learning, on the other hand, provides representations sensitive to specific image transformations while remaining invariant to others. By introducing equivariance to time-induced transformations, such as the anatomical changes in longitudinal medical images of a patient caused by disease progression, the model can effectively capture such changes in the representation space. However, learning temporally meaningful representations is challenging, as each patient's disease progresses at a different pace and manifests itself as different anatomical changes. In this work, we propose a Time-equivariant Contrastive Learning (TC) method. First, an encoder projects two unlabeled scans from different time points of the same patient to the representation space. Next, a temporal equivariance module is trained to predict the representation of a later visit based on the representation from one of the previous visits and from the time interval between them. Additionally, an invariance loss is applied to a projection of the representation space to encourage it to be robust to irrelevant image transformations such as translation, rotation, and noise. The representations learned with TC are not only sensitive to the progression of time but the temporal equivariant module can also be used to predict the representation for a given image at a future time-point. Our method has been evaluated on two longitudinal ophthalmic imaging datasets outperforming other state-of-the-art equivariant contrastive learning methods. Our method also showed a higher sensitivity to temporal ordering among the scans of each patient in comparison with the existing methods.

## 1 Introduction

The advent of contrastive pre-training methods showed that it is possible to learn informative and discriminative image representations by learning invariances to image transformations that do not alter their semantics. Their loss functions are calculated along the batch dimension and are specifically designed to produce discriminative features, avoiding trivial solutions such as representational collapse, where the network outputs the same representation regardless of the input. The learned invariances without the trivial solution to the transformations lead to more robust image representations, resulting in better generalization and performance on downstream tasks. However, sensitivity to some of these transformations may be crucial for specific downstream tasks, such as color information for flower classification or rotation for traffic sign detection. Thus, features with invariance/insensitivity to such relevant transformations result in degraded downstream task performance.

Transformation-sensitive (equivariant) contrastive methods are developed to learn sensitivity to a transformation in the representation space. An equivariant representation space should have a smooth and measurable response to the transformation applied in the image space. Unlike methods with specific architecture or layers (Cohen & Welling, 2016) for equivariance, the equivariant contrastive learning methods are mostly architecture agnostic. Instead, equivariance is achieved by recovering the image transformation parameter (Dangovski et al., 2022; Lee et al., 2021), or predicting the transformed representation in a projection space (Garrido et al., 2023; Devillers & Lefort, 2023). However, parameter recovery does not guarantee an equivariant representation space. Concurrently, learning perfect invariance is a trivial solution for learning to transform the represen-

tations, where the representation transformation is just an identity mapping. It can be prevented by either using the transformation function from the symmetry group ($SO(3)$ for rotation) in the image space or by calculating another contrastive loss for the transformed representations.

Introducing equivariance to time-induced transformations, such as anatomical changes due to disease progression in longitudinal medical images of a patient could be pivotal in downstream tasks such as predicting the future risk of conversion of a patient from an early to a late-stage of the disease. Degenerative eye diseases such as Age-related Macular Degeneration (AMD) and Glaucoma lead to irreversible tissue damage in the retina, implying that the severity of the disease can only increase, hence representing a monotonically non-decreasing function over time. However, the speed of disease progression and the associated anatomical changes occurring in the retina vary widely between patients and cannot be easily modeled or defined in the image space. Although longitudinal medical scans are routinely acquired in clinics to monitor disease progression, diagnostic labels for a desired prognostic task, e.g., time to conversion to late-stage disease, are often unavailable and expensive to obtain due to the medical expertise required to perform manual labeling. Consequently, a longitudinal Time-Equivariant SSL (self-supervised learning) pre-training step could be crucial in identifying the patients at a higher risk of progression to an advanced disease stage in order to provide timely and personalized treatment.

In this work, we proposed to learn time-sensitive/equivariant representations within a contrastive learning setting by generating a displacement map from an initial time point with a learnable transformation parameterized by temporal difference, which moves the representation in time. Since degenerative diseases only progress or, at most, stagnate over time, we modeled the disease progression in the representation space as an additive displacement map parameterized by the normalized time difference. Unlike the other transformed representation prediction methods (Garrido et al., 2023; Devillers & Lefort, 2023), we do not need to access the image transformation function or make any assumptions about it, since the transformed image is already available in the temporal dataset. Furthermore, we implemented the equivariance property directly in the representation space rather than in a projection space, allowing explicit manipulation of the representations. We adapted a regularization term on the calculated additive term to prevent the network from ignoring the time difference parameter and producing null displacement maps without relying on a second contrastive task, hence reducing complexity and computational overhead. We investigated the quality of representations and displacement maps by testing them on multiple temporal disease progression datasets with regular and irregular time intervals. The network was able to correctly order multiple scans of a patient, even though it was trained to order two time points with a limited time difference. In summary, our contributions are:

- We developed an equivariance module that can directly predict transformation in the image representation space for generating future representations without needing the corresponding patient scans.
- We introduced a regularization loss term on the norm of the predicted displacement map to prevent invariance to the time parameter.
- We constructed contrastive pairs from different visits of a patient for learning patient-specific features while enforcing temporal equivariance for features related to the disease progression.

## 2 RELATED WORKS

**Contrastive learning** as an unsupervised representation learning method, relies on learning invariance to unwanted image perturbations by increasing the representational similarity between the original image and its transformed versions in the representation space, while pushing other pairs apart. For the pulling-pushing operation, InfoNCE (Oord et al., 2018) loss is commonly used, which is calculated along the batch dimension. Ideally, the network should not produce constant representations disregarding the input, a trivial solution called *representational collapse*. The collapse can be prevented by using very large batch sizes for effective pulling and pushing (Chen et al., 2020), or replacing one of the input branches with a momentum encoder (He et al., 2020). As an alternative approach, non-contrastive methods do not rely on large batch sizes to prevent collapse since they do not define explicit negatives. Instead, they prevent representational collapse either by creating discrepancy between the branches with different techniques, such as predicting the output of the

other branch (Grill et al., 2020), applying a stop-gradient operation (Chen & He, 2021), or using a loss function to increase the correlation within pairs (Bardes et al., 2022; Zbontar et al., 2021), while decorrelating the rest. Moreover, the loss functions used in Bardes et al. (2022); Zbontar et al. (2021) prevent another type of collapse, coined as *informational collapse* where most of the vector dimensions become uninformative (Jing et al., 2022).

**Equivariant self-supervised learning** aims to learn representations sensitive to certain image transformations, unlike contrastive methods that seek total invariance. Indeed, invariance to certain transformations is not always desirable (Xiao et al., 2020). For instance, color contains discriminative information about the types of flowers. This sensitivity can be achieved with pretext self-supervised learning, by learning to recover the parameter of the target transformation from the representations, thus ensuring that the learned representations are sensitive to the parameter. Image rotation prediction (Gidaris et al., 2018) for rotation equivariance, video clip ordering (Xu et al., 2019) and distinguishing close and far clips (Jayaraman & Grauman, 2016) for temporal equivariance can be given as examples. In medical imaging, the time difference or ordering prediction between two visits (scans) of a patient is commonly used as a pre-training step for temporal tasks such as disease progression (Rivail et al., 2019). A network can order scans correctly only if it learns to recognize irreversible changes in the anatomy with the passage of time (Kim & Sabuncu, 2023). Zhao et al. (2021) decoupled the aging effect from disease-related changes in brain MRI scans by carefully curating a temporal dataset consisting of old patients without any underlying diseases. Then, they learned an aging trajectory in the representation space shared by all patients. Although these methods learn sensitivity to the transformation, they lack the desired invariance to image perturbations.

**Equivariant contrastive learning** has the benefit of learning equivariance to relevant transformations and invariance to the rest. Dangovski et al. (2022) combined transformation parameter prediction and contrastive loss for the transformation invariance. It requires 6 shared encoders; 2 for the contrastive task and 4 for the rotation predicting task (Gidaris et al., 2018), increasing the computational overhead substantially. Lee et al. (2021) enforced equivariance on the representation space by predicting the transformation parameter between the input pair, then projected these representations to an invariant space for the contrastive task. However, the ability to recover transformation parameters does not optimize and guarantee explicit equivariance, because the transformation parameter can be contained in a single dimension of representation, while the other dimensions could be totally invariant. Also, parameter prediction models do not learn any explicit function for predicting the effect of the target transformation on the representations. As a solution to both, Devillers & Lefort (2023) proposed to utilize two different projections of the representation space; one for the invariance task, and the second for the equivariance task. The equivariance branch has a trained module that predicts the transformed representation by conditioning on the untransformed representation and the image transformation parameter, allowing direct manipulation of the projections. Since invariance is a trivial solution for the equivariance where the equivariant representation transformation collapses to identity, they introduced additional contrastive loss for the equivariance branch. Later on, Garrido et al. (2023) decoupled the transformation of the representations from the content (*representations*), similarly image rotation matrix does not depend on the image content by directly employing the definition of transformation (i.e., for $SO(3)$ rotation transformation, $Sp(1)$ is used). The representation space is linearly split into two for equivariance and invariance. In the equivariance branch, a predictor conditioned only on the image rotation degree predicts the parameters for $Sp(1)$ representation transformation, and then the transformation is applied directly to the representations to rotate them. An extra contrastive loss is added for the rotated representations, similar to Devillers & Lefort (2023) with an increased computational overhead.

## 3 METHOD: TEMPORALLY EQUIVARIANT CONTRASTIVE LEARNING

### 3.1 TEMPORAL EQUIVARIANT REPRESENTATION

The progression of degenerative disease in a patient is routinely monitored using a series of medical image scans $x \in \mathbb{I} \subseteq \mathbb{R}^{N \times N}$ where $\mathbb{I}$ acquired across multiple visits, and $x_t$ denotes the scan acquired at a time $t$ sampled from $\mathbb{T} = \{t \in \mathbb{N}_0; t \leq b\}$, $b$ being the last visit date. Let $\mu : \mathbb{I} \times \mathbb{T} \to \mathbb{I}$ represent a transformation in the image space for all possible time differences, it captures the complex anatomical changes between $x_t$ and $x_{t+\Delta t}$ due to the disease progression (and/or normal aging), where $t + \Delta t \in \mathbb{T}$. Although the actual transformation $\mu$ is unknown and difficult to model

in the image space, the result of the transformation is available as $\boldsymbol{x}_{t+\Delta t}$ for some specific $\Delta t$, corresponding to available visits of the patient, such that $\boldsymbol{x}_{t+\Delta t} := \mu(\boldsymbol{x}_t, \Delta t)$. Let $f_\theta : \mathbb{I} \to \mathbb{R}^D$ with learnable parameters $\theta$ be a deep learning based encoder, such that it can embed $\boldsymbol{x}_t$ as a representation $r_t := f_\theta(\boldsymbol{x}_t)$ and $\boldsymbol{x}_{t+\Delta t}$ as $r_{t+\Delta t} := f_\theta(\boldsymbol{x}_{t+\Delta t})$. As $\mu(\cdot, \Delta t)$ transforms $\boldsymbol{x}_t$ to $\boldsymbol{x}_{t+\Delta t}$ in the image space, a corresponding transformation $h(\cdot, \Delta t)$ operates in the representation space such that $r_{t+\Delta t} = h(r_t, \Delta t)$. If $f_\theta$ is equivariant with respect to time, we can define the time equivariant relation as:

$$\forall \Delta t \in \mathbb{T}, \exists h \quad f_\theta(\mu(\boldsymbol{x}_t, \Delta t)) = f_\theta(\boldsymbol{x}_{t+\Delta t}) \approx h(f_\theta(\boldsymbol{x}_t), \Delta t) \tag{1}$$

A trivial solution for Eq. 1 would be *invariance* of $f_\theta$ such that $f_\theta(\boldsymbol{x}_t) = f_\theta(\boldsymbol{x}_{t+\Delta t})$ with disease progression being ignored, in turn making $h(\cdot, \Delta t)$ an identity mapping. This implies that $f_\theta$ learns only patient-specific features staying constant with respect to time while ignoring changes due to the progression. The transformation $h(\cdot, \Delta t)$ should be able to capture the future changes in the scans specific to each patient, which we propose to approximate with a deep neural network that can model complex changes.

## 3.2 TIME-EQUIVARIANT CONTRASTIVE LEARNING (TC)

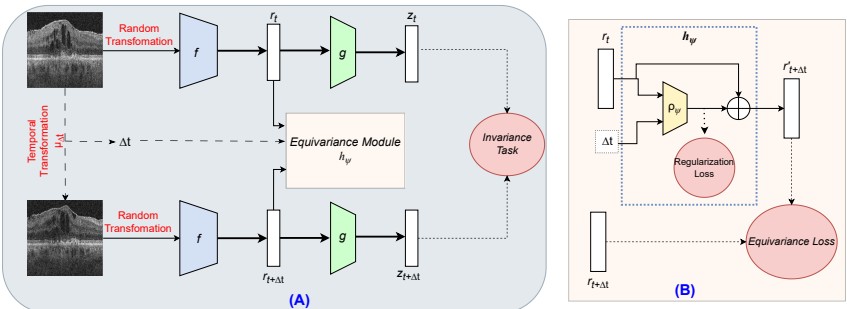

Figure 1: **(A)** End-to-end overview of **TC** pipeline. Representations from two-time points and their time difference are fed into the equivariance module for time equivariance, then projected to apply the invariance loss. **(B)** Equivariant transformation with an additive displacement map.

Our model builds upon the already existing invariant contrastive methods with a temporal equivariance. The input pair to the Siamese network is created from 2 different visits of a patient and both are transformed with contrastive augmentations (Fig.1.A). Recent contrastive equivariance methods with a learnable transformation (Devillers & Lefort, 2023; Garrido et al., 2023) on features, relied on 2 projection spaces, one for equivariance and the other for invariance, preventing them from directly steering the representations. Instead, we propose to enforce equivariance directly on the representation space, then to learn invariance on a *projection* space (Fig.1.A). This allows to preservation of the time-sensitive information in the representation space while enforcing invariance in the projection space. A logical order since invariance is a trivial solution of equivariance. It also simplifies the network architecture and computational overhead and facilitates transferring the encoder weights as a backbone. Calculating the contrastive loss in a projection space is a common practice in contrastive methods since it is helpful for the loss calculation (Jing et al., 2022).

We introduce an equivariance module $h_\psi$ containing the learned transformation (**predictor**) with learnable parameters $\psi$, which takes the concatenation of representation $r_t$ and normalized (between 0-1) time difference $\Delta t$ as input, and transforms $r_t$ to $r'_{t+\Delta t}$ to satisfy the equivariance property defined in Eq. 1. The predictor enables the model to generate future visits' representations by forwarding available scans in time to be used in predictive tasks. The equivariance loss for $h_\psi$ is defined as:

$$\ell_{\text{equiv}} = \|f_\theta(\boldsymbol{x}_{t+\Delta t}) - h_\psi(f_\theta(\boldsymbol{x}_t), \Delta t)\|_2^2 = \|r_{t+\Delta t} - h_\psi(r_t, \Delta t)\|_2^2 \tag{2}$$

As it is highlighted in Devillers & Lefort (2023); Garrido et al. (2023), the contrastive loss imposes invariance to the image transformations including time-related transformations. The learned invari-

ance results in a trivial solution for the equivariance loss where $h_\psi$ collapses to an identity mapping, ignoring $\Delta t$. The aforementioned methods rely on another set of computationally heavy contrastive task on $r'_{t+\Delta t}$ to prevent collapse. In the case of an MLP $h_\psi$, collapse happens if $\frac{\partial h_\psi}{\partial t}$ becomes null. If it is a single linear layer, the collapse manifests as the weights for $\Delta t$ being zero. In order to avoid additional contrastive task and a regularization term on the gradients to prevent $h_\psi$ from collapsing, we reparametrized $h_\psi$ as an additive displacement term:

$$r'_{t+\Delta t} = h_\psi(r_t, \Delta t) = r_t \oplus \rho_\psi(r_t, \Delta t) \tag{3}$$

Now, the equivariant transformation $h_\psi$ has two parts: an MLP $\rho_\psi$ for predicting displacement map as an additive term and *direct sum* of the displacement map with the representation (Fig. 1.B). In Eq. 3, the collapse condition is much easier to regularize since it is zero norm for the predicted displacement map. The ideal regularization should prevent displacement norm getting zero, while not encouraging high norm values. Because the representation space should respond to the time change smoothly without predicting large displacement maps easily, especially in the context of age-related degenerative diseases. We adapted a regularization loss term for preventing collapse from pair-wise loss of RankNet (Burges et al., 2005). The original loss function aims to increase the probability of ranking two embeddings correctly. In our case, the ranking is always one-sided, because the displacement norm cannot take negative values. Thus, the final displacement map regularization term becomes:

$$\ell_{\text{reg}} = \log\left(1 + \exp(\frac{-||\rho_\psi(r_t, \Delta t)||_2}{\tau})\right) \tag{4}$$

The $\ell_{\text{reg}}$ penalizes null displacements without encouraging very high norms. The temperature hyperparameter $\tau > 0$ affects the sensitivity of Eq. 4 to higher norm values.

For invariant contrastive learning, we used VICReg (Bardes et al., 2022) for its compatibility with TC. The contrastive loss aims to learn total invariance to image level augmentations, such as rotation, color jittering, blurring, and time-related changes. It is calculated using projections $z_t$ and $z_{t+\Delta t}$, obtained by projecting representations $r_t$ and $r_{t+\Delta t}$ with an MLP projector $g_\gamma$. Unlike $\ell_{\text{equiv}}$ 2 and $\ell_{\text{reg}}$ 4, it is calculated along the batch, where a batch of projections is defined as $Z_t$ and $Z_{t+\Delta t}$, where each pair could have different $\Delta t$ values. VICReg contrastive loss with *invariance*, *variance*, and *covariance* terms, is defined in order as:

$$\ell_{\text{contr}}(Z_t, Z_{t+\Delta t}) = \lambda_S \cdot S(Z_t, Z_{t+\Delta t}) + \lambda_V \cdot (V(Z_t) + V(Z_{t+\Delta t})) \tag{5}$$
$$+ \lambda_C \cdot (C(Z_t) + C(Z_{t+\Delta t}))$$
$$S(Z, Z') = \frac{1}{n}\sum_i \|z_i - z'_i\|_2^2$$
$$V(Z) = \frac{1}{d}\sum_{j=1}^{d} max(0, 1 - \text{std}(z^j, \epsilon))$$
$$C(Z) = \frac{1}{d}\sum_{i \neq j}[Cov(Z)]_{i,j}^2,$$
$$\text{where} \quad Cov(Z) = \frac{1}{n-1}\sum_{i=1}^{n}(z_i - \bar{z})(z_i - \bar{z})^T.$$

Thus, the final training loss function is:

$$\mathcal{L}_{\text{TC}} = \ell_{\text{contr}} + \beta \cdot (\ell_{\text{equiv}} + \upsilon \cdot \ell_{\text{reg}}) \tag{6}$$

## 4 EXPERIMENTS

We tested **TC** against 3 equivariant contrastive methods: ESSL (Dangovski et al., 2022), Aug-Self (Lee et al., 2021) and EquiMod (Devillers & Lefort, 2023), and VICReg was used as a baseline without equivariance. During pre-training each batch contains only a single image pair from a patient to avoid pushing apart the intra-patient scans with the contrastive loss. Since, time-related changes such as the loss of tissue are irreversible and asymmetric, it is difficult to model the transformations backward in time. Moreover, in clinical applications, only future representations are required for assessing the disease severity and planning the drug administration in a timely manner. Therefore, the model was pre-trained only with training image pairs with a positive $\Delta t$.

### 4.1 DATASETS

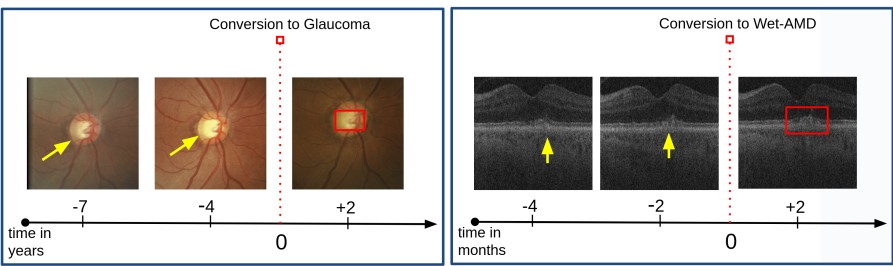

Figure 2: Examples of longitudinal images acquired across multiple visits of a patient showing signs of disease progression; **Left**: Color Fundus images with conversion to Glaucoma (SIGF dataset); **Right**: OCT scans with conversion to wet-AMD (HARBOR dataset).

We evaluated our method on two longitudinal medical datasets for degenerative disease progression, a private HARBOR dataset of retinal Optical Coherence Tomography (OCT) images acquired at regular time intervals and a publicly available SIGF dataset of color fundus images which were acquired at irregular time intervals (Fig. 2).

**HARBOR Dataset:** It is a dataset from the longitudinal AMD study consisting of 3D OCT scans for *fellow-eyes*[1] of patients undergoing treatment for wet-AMD. The fellow-eye scans show retinas in the early or intermediate AMD stage and contain cases that irreversibly convert to the late stage of wet-AMD (Ho et al., 2014). The dataset consists of monthly follow-up scans over a 2-year period, yielding 24 scans per eye. We used scans of eyes in the intermediate AMD stage to predict the onset of wet-AMD within 6 months as a binary downstream task from a single visit. Out of 463 fellow eyes (10,108 OCT scans) with initial intermediate-stage AMD diagnosis, 117 of them have an observed conversion to wet-AMD. The rest of the 540 fellow eyes (12,506 scans) are only used in the pre-training step. For computational efficiency, we extracted the central 2D slice from each OCT volume passing through the fovea (the retinal region responsible for sharp central vision) which is known to be most representative of the disease state (Lin et al., 2021). The images were resized to a $224 \times 224$ gray-scale image with intensities normalized to $[0, 1]$.

We followed Emre et al. (2022) for contrastive image transformations specific to B-scans. For the equivariant contrastive training, we limited $\Delta t$ between 1 to 9 **months** and normalized it between $0 - 1$. The rationale behind this is that the set of all pair permutations for all possible time differences is very large (276 per patient) for a plausible training time. By limiting $\Delta t$, training $h_\psi$ becomes much more convenient, because it only needs to produce displacement maps for a constrained but practically meaningful interval.

**SIGF Dataset:** It is a public longitudinal dataset of color **Fundus** images showing progression to *Glaucoma* (Li et al., 2020). Glaucoma is an irreversible disease affecting the optic nerve and can be detected using Fundus images, which are color photographs of the back of the eye that capture the retina and the optic disc. In the SIGF dataset, there are 405 eyes with 9 Fundus images

---

[1]The other eye that is not part of the drug trial study

Table 1: Predictive performance of the evaluated models on **HARBOR** and **SIGF** datasets.

| Model | HARBOR | | | SIGF | |
|---|---|---|---|---|---|
| | AUROC $\uparrow$ | PRAUC $\uparrow$ | CIp $\uparrow$ | AUROC $\uparrow$ | CIp $\uparrow$ |
| VICReg | $0.69 \pm 0.01$ | $0.13 \pm 0.02$ | 0.56 | 0.73 | 0.64 |
| ESSL | $0.70 \pm 0.04$ | $0.13 \pm 0.01$ | 0.62 | 0.74 | 0.62 |
| AugSelf | $0.70 \pm 0.03$ | $0.14 \pm 0.02$ | 0.61 | 0.80 | 0.67 |
| EquiMod | $0.72 \pm 0.02$ | $0.14 \pm 0.02$ | 0.59 | 0.78 | 0.66 |
| TC w/o *DM* | $0.71 \pm 0.03$ | $0.12 \pm 0.01$ | 0.66 | 0.79 | 0.68 |
| TC w/o $\ell_{\text{reg}}$ | $0.72 \pm 0.03$ | $0.13 \pm 0.01$ | 0.66 | 0.82 | 0.66 |
| TC | $\mathbf{0.74 \pm 0.02}$ | $\mathbf{0.16 \pm 0.03}$ | $\mathbf{0.68}$ | $\mathbf{0.85}$ | $\mathbf{0.69}$ |

per eye spanning 32 years. Each scan has dimensions of $3 \times 224 \times 224$. For the downstream task of predicting the conversion to early Glaucoma and the associated data splits, we followed the definitions in Li et al. (2020). After determining the possible Fundus perturbations with expert ophthalmologists, we chose Gaussian blurring, random cropping, solarization, translation, and small rotation for contrastive augmentations. Also, all right eyes are mirrored to look like left eyes since the eye position is readily available, and there is no need to learn invariance for it. During training, $\Delta t$ is constrained between $1 - 24$ *years*.

## 4.2 SETTING

We used ResNet50 with a representation dimension of 2048 for HARBOR as the encoder backbone, and ResNet18 for SIGF due to the smaller dataset size. Following the common practice of projecting representations to higher dimensions (Bardes et al., 2022; Zbontar et al., 2021), the projector $g_\gamma$ is a three-layer MLP with dimensions 4096 and 1024 for ResNet50 and Resnet18 respectively. Displacement map predictor $\rho_\psi$ of TC is implemented as a two-layer MLP with an input size of $(2048/512 + 1)$ for $[r_t, \Delta t]$, and it outputs a displacement map with the same dimension as the representations. In EquiMod and AugSelf, we followed the original works for the predictor architecture. Original ESSL investigates rotation equivariance by predicting 0, 90, 180, and 270-degree rotations. In contrast, time difference prediction is relative, requiring two time points to make a prediction. Thus, we concatenated two representations and fed them to a regression head to predict the time difference. It means we reduced the number of encoders to 4 from 6 in the ESSL setting.

All of the models are pretrained for 300 epochs with the AdamW optimizer. Since there is only a single pair per patient in a batch, an epoch is extended by 25 times for HARBOR and 15 times for SIGF by resampling random pairs from patients. The learning set is set to 5e-3 for HARBOR and 5e-4 for SIGF correlated with the dataset sizes. A cosine scheduler with a warm-up is used for decaying learning rates. The same weight decay of 1e-6 is applied to all pre-training setups. Batch size is set to 256 and 128 for Harbor and SIGF. $\beta$ and $\upsilon$ of $\mathcal{L}_{\text{TC}}$ are set to 0.5 and 0.1 respectively. Contrastive VICReg loss hyperparameters $\lambda_S$, $\lambda_V$, and $\lambda_C$ are set to 15, 25, 5 for HARBOR and 25, 25, 5 for SIGF to bring the magnitude of the loss terms in the same range.

## 4.3 RESULTS

### 4.3.1 LINEAR EVALUATION OF DOWNSTREAM PREDICTION PERFORMANCE

After pre-training, we evaluated the extracted representations with a linear classifier on the downstream tasks described in Sec. 4.1. The classifier is trained for 50 epochs with Adam optimizer and a learning rate of 1e-4. We reported the results with Area Under the Receiver Operating Characteristic (AUROC) due to a large class imbalance in both datasets (1:20 for both).

The performance of the Linear evaluation is presented in Table 1. In terms of AUROC, our TC outperforms all other methods on both datasets. Only EquiMod achieved a comparable AUROC on the HARBOR dataset. But TC is more computationally efficient and offers a direct manipulation of the representations. Especially when tested on the SIGF dataset, our TC achieves a remarkable 0.85 AUROC with a high margin to the closest AugSelf method (0.8 AUROC). The results confirm that

Table 2: Effect of temperature $\tau$ on the performance in HARBOR dataset.

| $\tau$ | AUROC | CIp |
|-----|-------|------|
| 0.5 | 0.75  | 0.67 |
| 1.0 | 0.77  | 0.68 |
| 2.0 | 0.74  | 0.60 |

the time-sensitive representations acquired with TC have better discrimination quality compared to the other equivariant models for the temporal disease progression tasks.

### 4.3.2 EQUIVARIANCE EVALUATION OF THE LEARNED REPRESENTATIONS

We evaluate the temporal equivariance property of the learned representation using the Concordance Index (Harrell et al., 1982) computed individually for each patient and then averaged across all patients (CIp) in the test set. For each patient, the euclidean distance between a baseline scan (initial visit of a patient) representation and all other scan representation for future visits is used as a measurement for ranking. Ideally, the distance should monotonically increase with time, CIp quantifies the equivariance property with the representational distance between the baseline visit and other visits of a patient for which the ranking metric predicts the correct order. A CIp value of 1.0 highlights that representations of a patient are on a smooth trajectory (monotonically increasing distance with time) resulting from a strong time equivariance property of the representation space, while 0.5 means random ordering without any discernable correlation with time. Finally, when the order is completely reversed (monotonically decreasing distance with time) CIp is 0.

Representations from TC achieve higher CIp values, highlighting a more smooth trajectory over time. When TC is compared against EquiMod, they achieve a similar AUROC score, but TC outperforms it significantly in terms of CIp, highlighting its strong equivariance property. For the SIGF dataset, the differences in CIp between the methods are small, which we attribute to the changes related to Glaucoma being subtle on Fundus images.

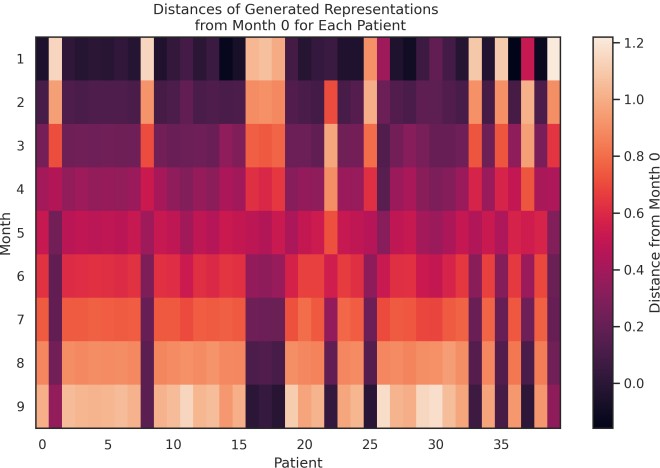

Figure 3: Distance between original represnentation $r_0$ and its equivariant prediction $r'_{t+\Delta t}$ rankings for synthetically created representations.

We re-created the HARBOR test set by transforming each patient's initial visit representations in time for consecutive 9 visits. $h_\phi$ enables us to obtain future representations directly. The resulting synthetic dataset has a CIp of 0.77, higher than the reported value of 0.68. This shows that $h_\phi$ manages to create a straight trajectory in the representation space only using an initial point. Inter-

estingly, CIp is either close to 1.0 (perfect ordering - most of the cases) or close to 0 (perfect reverse order) but almost never close to 0.5 (random order) (Fig. 3).

### 4.3.3 ABLATION STUDIES

We investigated the impact of the explicit displacement map prediction (Table 1 - TC w/o *DM*) and the regularization loss term (Table 1 - TC w/o $\ell_{reg}$) components of TC by removing each at the time. In both cases, the downstream task performance degraded considerably on both of the datasets, highlighting their importance. This drop in AUROC can be attributed to less discriminative representations. On the other hand, the decrease in CIp values is minimal. Considering the two metrics jointly, the ablation experiments signify that TC is still capable of learning temporal equivariance even with lower representation quality (low AUROC). When TC w/o $\ell_{reg}$ compared against TC w/o *DM*, it performs slightly better. This can be attributed to our reformulation with displacement maps. The collapse due to total invariance manifests in the predicted displacement maps rather than the representations.

High values for temperature $\tau$ of $\ell_{reg}$ in Eq. 4 increase $\ell_{reg}$'s sensitivity to displacement maps with high norm values, in turn leading to even larger norm values for optimizing $\ell_{reg}$. While the term should prevent zero norm, it should not encourage higher norms in order to prevent sudden changes when predicting future representations. In Table 2, we found that smaller temperature values provide better CIp values, preventing representations from being very far apart and losing the temporal equivariance property over long periods.

## 5 CONCLUSION

Current equivariant contrastive methods aim to achieve sensitivity to certain image transformations. They rely on transformation parameter recovery, or learning a mapping in a projection space such that information related to desired transformations, is preserved in the representations, in contrast with common contrastive methods. In this paper, we proposed TC which has a learnable equivariance module for directly transforming the medical image representations in time. Proposed temporal transformation retains the characteristics of disease progression over time such as irreversible loss of tissue or accumulating damage in anatomy over time. We introduced a regularization loss term that prevents trivial solution for the equivariance induced by the contrastive loss, unlike other equivariant models that rely on an additional contrastive task increasing the computational complexity.

We tested TC by linear evaluating for disease progression on two temporal datasets consisting of eye scans (Fundus and OCT) with different degenerative diseases. It always improved the performance against the other equivariant contrastive methods. The results highlight the importance of temporal sensitivity in the representations in order to correctly assess patients' risk of conversion to a late disease stage. Additionally, we observed that the synthetically created representations for future visits follow a smoother trajectory analogous to the expected progression of degenerative diseases.

Temporal medical datasets are costly to obtain, and very difficult to release publicly. As a future work, we plan to test our representation transformation feature for expanding existing temporal datasets. This could improve predictive performance, especially in the case of time-series models that benefit from images with regular intervals. Similarly, another research line could be bringing in the equivariance property to the representation from an already pre-trained model by combining it with an equivariance module. Then, equivariant representations could be obtained without pre-training. The major limitation of TC is that it relies on irreversible diseases with scans acquired in discreet intervals. Thus, we are planning to improve the equivariance module to make the predicted transformation reversible, similar to rotating an image back and forth.

## 6 REPRODUCIBILITY STATEMENT

We provided detailed preprocessing and the contrastive augmentations in Section A. SIGF is a public dataset available to download, we will release the code publicly for its pretraining, linear evaluation and CIp testing 4.3.2 steps.

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

## A  APPENDIX

### A.1  ADDITIONAL DETAILS ON THE DATASET

The HARBOR dataset consists of OCT scans acquired with a Cirrus OCT scanner. Each scan covers an area of $6 \times 6 \times 2$ mm$^3$. As a pre-processing step, the retina in each B-scan is *flattened* with respect to the innermost layer called *Bruch's Membrane*. Then the resulting B-scan is cropped between Bruch's Membrane and the outermost layer and resized to $224 \times 224$ with its pixel values normalized between $0-1$. Even though wet-AMD is a degenerative and irreversible disease, there is an intravitreal injection that mitigates the patient's symptoms. However, it is only effective when it is applied right after the wet-AMD conversion occurs. Thus, it is necessary to predict the conversion beforehand to plan a timely drug administration. Thus, we chose wet-AMD conversion prediction within a time window of 6 months.

The SIGF contains colorfundus images with irregular time intervals. The average interval is 1.67 years with a standard deviation of 1.34. It is important to note that the largest interval is 13 years.

In Table 3, we detailed the contrastive augmentations applied to each dataset.

### A.2  DIRECT TRANSFORMATION OF REPRESENTATIONS

In Fig. 4, the representation $r'_{t+\Delta t}$ is directly predicted by $h_\psi$ without calculating the displacement map. We evaluated it by replacing the equivariance module with it in Sec. 4.3.3 as ablation studies.

Table 3: Contrastive Augmentations

| Augmentation | HARBOR | SIGF |
|---|---|---|
| Random Crop & Resize (percentage) | 0.4 - 0.8 | 0.7 - 0.9 |
| Random Horizontal Flip (probability) | 0.5 | 0 |
| Random Color Jittering (probability | 0.8 | 0.8 |
| Random Gaussian Blur (kernel size) | 21 | 21 |
| Random Solarize (threshold) | 0.42 | 0.9 |
| Random Rotation (degrees) | − | ±25 |
| Random Translation (percentage) | ±0.05 | ±0.05 |
| Input Time Difference | 1-9 Months | 1-24 Years |

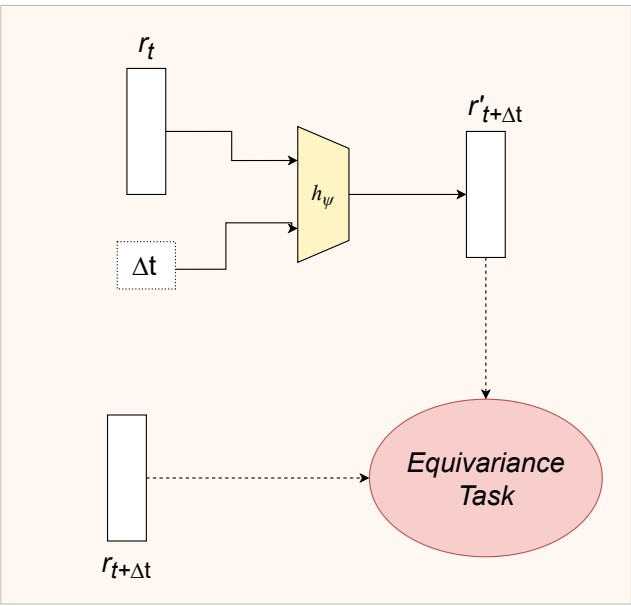

Figure 4: Equivariant transformation by predicting the transformed representation directly.

## A.3 TEMPERATURE $\tau$ OF $\ell_{\text{REG}}$

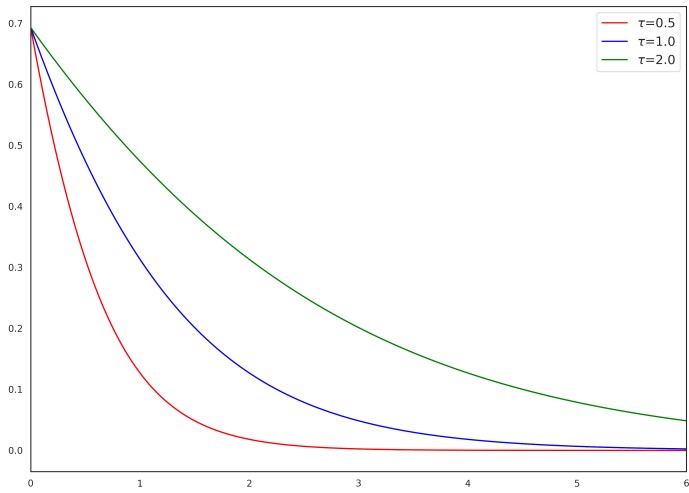

Figure 5: Different sensitivity of $\ell_{\text{reg}}$ to norm values with different $\tau$

## A.4 COMPUTATIONAL COSTS

In Table 4, we provided average batch update time and total number of trainable parameters for TC and the other methods. Compared to EquiMod, another method for predicting the transformed representations, TC has substantially fewer parameters due to the fact that TC predicts the next time step in the representation space rather than a high dimensional projection space (Sec. 3.2). Additionally EquiMod relies on additional contrastive task to prevent trivial equivariant solution, contributing to its higher traning time, unlike TC. The models relying on transformation parameter recovery, ESSL and AugSelf, have fewer parameter compared to TC and EquiMod, since their predictor outputs a scalar rather than a representation vector, but they have worse execution time because they both require additional transformed images to predict the transformation parameter. Consideting most of the medical datasets consist of volumes the efficiency of TC in terms of trainable parameters and time, makes it more scalable to higher dimensional datasets.

Table 4: Computational Costs of Each Model for the HARBOR Trial Dataset

| Model | Batch Update (Seconds) | # Trainable Parameters |
|---|---|---|
| VICReg | 9.3 | 65.4M |
| ESSL | 16.3 | 78.0M |
| AugSelf | 14.2 | 78.0M |
| EquiMod | 12.6 | 124.2M |
| TC | 10.4 | 82.2M |

## A.5 DERIVATION OF DISPLACEMENT MAP REGULARIZATION TERM

The pairwise RankNet (Burges et al., 2005) loss term is based on cross-entropy loss and is used for calculating the score-based ranking a pair of elements. Let $f$ be a scoring function for the items $x_i$

and $x_j$, such that their respective scores are $s_i = f(x_i)$ and $s_j = f(x_j)$. Concordantly the score difference is defined as $s_{ij} = s_i - s_j$. Then, the probability of ranking $x_i$ greater than $x_j$ is defined using the logistic function:

$$\mathcal{P}_{ij} = \frac{e^{s_{ij}}}{1 + e^{s_{ij}}} = \frac{1}{1 + e^{-s_{ij}}} \tag{7}$$

When ranking the items, there are 3 possible values for ground truth $Y$; 1 when $x_i$ has higher rank than $x_j$, 0 when the relation is reversed, and $\frac{1}{2}$ when both items have the same rank. Given this relation, cross-entropy loss for correctly ranking the items is given as:

$$\mathcal{L}_{ce} = -Y \cdot \log(\mathcal{P}_{ij}) - (1 - Y) \cdot \log(1 - \mathcal{P}_{ij}) \tag{8}$$

In terms of our problem, $s_i$ becomes $r_{t+\Delta t}$ and $s_j$ becomes $r_t$, hence $s_{ij}$ is the displacement map. For the cross-entropy loss, we implemented $s_{ij}$ as the norm of the displacement map. In TC, the ranking class is 1 because the time difference between the branches of the Siamese network is positive, and the cross-entropy loss becomes:

$$\mathcal{L}_{reg} = -1 \cdot \log(\mathcal{P}_{ij}) - (0) \cdot \log(1 - \mathcal{P}_{ij}) = -\log\left(\frac{1}{1 + e^{-s_{ij}}}\right) \tag{9}$$

$$= \log(1 + e^{-s_{ij}}) = \log\left(1 + \exp(\frac{-||\rho_\psi(r_t, \Delta t)||_2}{\tau})\right)$$

