# OpenReview forum: "Temporally Equivariant Contrastive Learning for Disease Progression"
_ICLR.cc/2024/Conference — ICLR 2024 Conference Withdrawn Submission_

### Official Review · Reviewer_mWX7 · 2023-10-31

**Soundness:** 3 good
**Presentation:** 3 good
**Contribution:** 2 fair
**Rating:** 5
**Confidence:** 4

**Summary:**

To incorporate the element of time into image transformation, the researchers introduced the Time-equivariant Contrastive Learning (TC) technique. This method involves training a temporal equivariance module, ensuring that the model captures representations across various timesteps. As a result, the model gains the capability to anticipate future image representations within a consistent time frame. Empirical evaluations conducted on two image datasets have demonstrated the effectiveness and robust performance of the proposed approach.

**Strengths:**

+ The employment of an equivariance module allows the model to directly predict the transformation of representations in correlation with image transformations, facilitating the generation of future image representations.
+ A regularization loss term has been incorporated, applied to the predicted displacement map, ensuring the time parameter invariance is upheld.
+ By constructing contrastive pairs for each patient visit, the dataset is enriched, promoting smooth training of the model.

**Weaknesses:**

-  Regarding equation 4, the regularization term: Would utilizing a normal L2 loss on the MLP, adjusted by a coefficient τ, be feasible? The rationale behind adding a constant 1 in this equation is unclear.
- The structure of ψ in equation 2 needs elaboration. Given its crucial role in predicting future representations, it seems unlikely to be just an MLP.
- Additional experiment results showcasing the impact of different loss item coefficients are necessary. Solely omitting one does not provide a thorough comparison.
- It would be beneficial to display some prediction results based on past image representations, possibly comparing actual images from a patient’s last visit to predicted ones over a set time period.
- Would utilizing negative time in training, aiming to predict previous images given a current one, contribute to enlarging the dataset?

**Questions:**

- Could you clarify the definition of 'time' in this context? In scenarios where a healthy individual undergoes multiple scans, the images might appear identical, whereas for a patient, variations are expected. This could potentially lead to the collapse of the time equivariance module with limited patient data.
- Is there a scope for the model to handle both 3D scans and 2D images as inputs? This might enable the model to learn a richer set of representations.
- Could you please explain the term "CIn testing" mentioned in section 6? It appears to be a novel concept without a provided definition.

---

> ### Author Response · Authors · 2023-11-23
>
> We would like to thank you for your  on-point review. Your comments will help us improve the submission.
>
> > Regarding equation 4, the regularization term: Would utilizing a normal L2 loss on the MLP, adjusted by a coefficient τ, be feasible? The rationale behind adding a constant 1 in this equation is unclear.
>
> The problem with L2 loss is that it does not prevent the collapse solution which manifests as 0 norm of the displacement map. That is true that it will prevent large displacement maps, but our main concern is to prevent the collapse (manifests as 0 norm displacement map) and L2 promotes the collapse solution by encouraging 0 norm displacement map. We added a derivation of the regularization term from the pairwise ranking loss in Appendix 4. Also we would like to highlight that without the adding constant 1, the loss term is 0 when the displacement norm is 0, which is exactly the opposite of its intention. And it would promote very large displacement map norms, because $\lim_{x \to \infty} \log\left(e^{-\frac{x}{\tau}}\right)$ is obviously -inf, whereas $\lim_{x \to \infty} \log\left(1+e^{-\frac{-x}{\tau}}\right)$   is 0.
>
> >The structure of ψ in equation 2 needs elaboration. Given its crucial role in predicting future representations, it seems unlikely to be just an MLP.
>
> $h_ψ$ is the temporal representation prediction module, which include a displacement map prediction 2 layer MLP $p_ψ$. Although a more elaborate prediction network would improve the results, such as a very recent work [Garrido, Q., et al. (2023) Self-supervised learning of Split Invariant Equivariant representations.], we followed the MLP architecture described in Equimod and AugSelf for predicting the representation from the Siamese Network in order to highlight our reformulation of the equivariance task. Thank you for highlighting the shortcomings of the architecture, we are planning to build a more specific network for the equivariance prediction as a future step.
>
> >Would utilizing negative time in training, aiming to predict previous images given a current one, contribute to enlarging the dataset?
>
> With our current time interval selection (30-270 days), the HARBOR dataset yields 180 pairs per patient from 24 visits. Additionally, in terms of clinical application, it is more interesting to predict only the future, such as for planning future visits. Given additional computational overhead, and the extra complexity from trying to reverse a non-linear transformation, we believe such dataset enlargement would not be beneficial
>
> >Could you clarify the definition of 'time' in this context? In scenarios where a healthy individual undergoes multiple scans, the images might appear identical, whereas for a patient, variations are expected. This could potentially lead to the collapse of the time equivariance module with limited patient data.
>
> Your concern is on the point and correct. In this study we aim to address the degenerative disease datasets which the disease generally progresses with varying degrees, and rarely stagnates. As you highlighted if there are healthy patients in the dataset, calculating displacement map would lead to trivial solutions. On the other hand, unsupervised split of HARBOR trial dataset includes late-stage wet-AMD patients. In this trial, those patients are treated with an intravitreal injection called anti-VEGF. It reduces  As a future work, we are planning to address reversed and null changes in time.
>
> >Is there a scope for the model to handle both 3D scans and 2D images as inputs? This might enable the model to learn a richer set of representations.
>
> Our internal experiments with 3D OCT scans achieve higher than what is reported with 2D slices (B-Scans). But OCT scans are highly dense with 128 B-Scans, which increase the computational substantially. Especially, EquiMod has 2 contrastive tasks, which makes it almost unfeasible to train. We could have fit only batch size of 8 (very low even for non-contrastive loss functions), and it took ~11 days to complete a single pretraining on a single Nvidia A100 80gb. Thus, we reverted back to using 2D retinal central B-Scan (called fovea), which is the most relevant position for Age-related Macular Degeneration [Andrew C Lin et al. Assessing the clinical utility of expanded macular octs using machine learning].
>
> >Could you please explain the term "CIn testing" mentioned in section 6? It appears to be a novel concept without a provided definition.
>
> Thanks for highlighting the phrase, it was a typo. It is changed to CIp as it is described in Section 4.3.2

---

### Official Review · Reviewer_m6pm · 2023-10-31

**Soundness:** 3 good
**Presentation:** 3 good
**Contribution:** 2 fair
**Rating:** 5
**Confidence:** 3

**Summary:**

The paper introduces a Time-equivariant Contrastive Learning (TC) method aimed at achieving temporally-sensitive representations within a contrastive learning setting. This is especially targeted at disease progression in longitudinal medical images. The method uses an encoder to project scans from different time points into a representation space and employs a temporal equivariance module that can predict future representations using existing ones. Unlike other methods, this approach directly transforms medical image representations in time, capturing irreversible anatomical changes, like those due to degenerative diseases. The authors introduced a regularization loss term to maintain the sensitivity of the time parameter and constructed contrastive pairs from different patient visits to learn patient-specific features. The method outperforms other equivariant contrastive methods on two temporal eye scan datasets, emphasizing the importance of temporal sensitivity for assessing disease progression risk. The TC method's main limitation is its reliance on irreversible diseases with scans acquired at discrete intervals.

**Strengths:**

S1. The paper presents an approach in the realm of contrastive learning. By introducing the Time-equivariant Contrastive Learning (TC) method, the authors tackle the challenge of capturing irreversible anatomical changes, particularly in the context of disease progression.

S2. The proposed TC method has been tested and has demonstrated superior performance against other state-of-the-art equivariant contrastive techniques. This is evident from its application on two temporal datasets related to eye scans, which solidifies its robustness and reliability.

S3. The paper does a good job of articulating the need and significance of the proposed method. The clear presentation of the challenges associated with degenerative diseases, especially the irreversible nature of certain anatomical changes, underscores the importance and timeliness of the TC method.

S4. The potential ramifications of the TC method in the medical field is highlighted. Its ability to predict disease progression and effectively assess a patient's risk of advancing to more severe stages of a disease can be a game-changer in patient care and medical diagnosis. This predictive capability is not just about identifying risk but also about enabling timely and personalized interventions.
S5. The TC method's ability to generate future representations without needing corresponding patient scans showcases its versatility. This feature is particularly beneficial for medical scenarios where timely scans might not be available but predictions are essential for patient care.

**Weaknesses:**

W1. While the TC method introduces a promising approach for handling irreversible diseases with discretely acquired scans, its utility seems confined to this specific scenario. The real world of medical conditions is vast, and many diseases do not follow a strictly irreversible path. The narrow scope potentially limits the method's broader applicability across the diverse landscape of medical imaging and conditions.

W2. The method heavily relies on the assumption that degenerative diseases follow a monotonic non-decreasing function over time. This is a strong assumption that might not hold across all conditions or datasets. Diseases can have varying trajectories, with periods of stability, rapid progression, or even temporary reversal. Basing a method on this assumption might lead to inaccuracies in real-world applications.

W3. Although the paper claims superior performance against other methods, a deeper comparative analysis, considering a broader range of conditions, datasets, and variability, would be more convincing. A comprehensive comparison would provide clarity on the margin and conditions of this outperformance.

W4. Medical images are notorious for their variability and potential anomalies, especially given the variation between patients, imaging equipment, and acquisition protocols. The paper does not delve deeply into how the TC method would handle potential anomalies, outliers, or inconsistencies in the longitudinal medical images.

W5. While the paper alludes to reduced complexity in comparison to other models, a more explicit discussion or breakdown of computational costs, resource requirements, and scalability considerations is conspicuously missing. For real-world application, especially in clinical settings, understanding computational overhead is crucial.

W6. The paper mentions that temporal medical datasets are costly to obtain and challenging to release publicly. This heavy reliance on hard-to-obtain datasets can be a significant bottleneck for the practical application and scalability of the TC method. Additional experiments with non-medical images would provide beneficial to applicability of this model into other fields and since it already has the potential, lack of its presence is a weakness in the paper.

W7. The introduction of a regularization loss term, crucial for maintaining time parameter sensitivity, is presented without an in-depth rationale or derivation. A clearer explanation of this term's derivation, its impact on the model's performance, and sensitivity analyses would provide more confidence in the method's robustness.

W8. There are only 3 baselines tested for the experiments and for these the performance improvement is minor. Error bars should be added to identify whether these performance improvements are significant and how robust to different random seeds is the model compared to the other baselines.

**Questions:**

Q1. How does the Time-equivariant Contrastive Learning (TC) method handle highly irregular intervals between patient visits?

Q2. How is the contribution of different loss terms in your overall total loss change throughout training? Do each component monotonically decrease? Additional analysis of these should be added to the appendix.

Q3. How does the model handle potential anomalies in the data, such as imaging artifacts or errors?

Q4. Have you considered extending the application of the TC method to other medical imaging modalities or even non-medical datasets?

Q5. What are the computational complexities involved in the TC method, especially when applied to large-scale datasets?

**Details Of Ethics Concerns:**

N/a.

---

> ### Author Response · Authors · 2023-11-23
>
> Thank your for your detailed reviews and comments.
> > W1. The paper presents an approach in the realm of contrastive learning. By introducing the Time-equivariant Contrastive Learning (TC) method, the authors tackle the challenge of capturing irreversible anatomical changes, particularly in the context of disease progression.
>
> In our setup, we targeted degenerative diseases that affect the retina. In AMD, the patient monitoring begins with a diagnosis of the intermediate stage. And the disease only progresses from that point. Although the application seems limited, AMD is the leading cause of blindness, affecting dozens of millions in the world. There is not much clinical insight on the biomarkers that cause the stage conversion. This makes good predictive models necessary for timely delivery of the available drugs and reducing the monitoring load of the hospitals, saving billions in health care cost.
>
> >W2. The method heavily relies on the assumption that degenerative diseases follow a monotonic non-decreasing function over time. This is a strong assumption that might not hold across all conditions or datasets. Diseases can have varying trajectories, with periods of stability, rapid progression, or even temporary reversal. Basing a method on this assumption might lead to inaccuracies in real-world applications.
>
> Unsupervised part of the Harbor dataset includes late-stage patients that are treated with anti-VEGF injection. The injection reverses some symptoms and reduces the fluid pockets. But it does not prevent the disease progression. We specifically did not remove those cases in order to have a more realistic set-up. In terms of optimization, they are similar to noisy samples (also there are other artifacts from erroneous scanning as well), and the models should be robust to it. The displacement map is conditioned on both the time difference and the current image, which is necessary to capture the individual trajectories and progression speeds. To make a comparison, the input image is not necessary to define to rotate the image. The 3x3 rotation matrix is enough. But in order to rotate an object within a 2D image, the rotation degree and the input image have to be provided.
>
> >W4. Medical images are notorious for their variability and potential anomalies, especially given the variation between patients, imaging equipment, and acquisition protocols. The paper does not delve deeply into how the TC method would handle potential anomalies, outliers, or inconsistencies in the longitudinal medical images.
>
> The Harbor dataset have scans after treatment creating temporal inconsistencies. We aim to extract the available temporal information from the unsupervised temporal dataset, hence the noise and the anomalies were expected. Similarly, we chose the SIGF as the second dataset, because the image acquisition intervals are highly irregular.
>
> >W5. While the paper alludes to reduced complexity in comparison to other models, a more explicit discussion or breakdown of computational costs, resource requirements, and scalability considerations is conspicuously missing. For real-world application, especially in clinical settings, understanding computational overhead is crucial.
>
> Compared to EquiMod, another method for predicting the transformed representations, TC has substantially fewer parameters due to the fact that TC predicts the next time step in the representation space rather than a high dimensional projection space. Additionally EquiMod relies on additional contrastive task to prevent trivial equivariant solution, contributing to its higher traning time, unlike TC. The models relying on transformation parameter recovery, ESSL and AugSelf, have fewer parameter compared to TC and EquiMod, since their predictor outputs a scalar rather than a representation vector, but they have worse execution time because they both require additional transformed images to predict the transformation parameter. Consideting most of the medical datasets consist of volumes the efficiency of TC in terms of trainable parameters and time, makes it more scalable to higher dimensional datasets.
> We added this discussion and the cost table in Appendix A.4.
>
> >W7. The introduction of a regularization loss term, crucial for maintaining time parameter sensitivity, is presented without an in-depth rationale or derivation. A clearer explanation of this term's derivation, its impact on the model's performance, and sensitivity analyses would provide more confidence in the method's robustness.
>
> We added the full derivation of the term in Appendix A.5.

---

> ### Author Response · Authors · 2023-11-23
>
> >W8. There are only 3 baselines tested for the experiments and for these the performance improvement is minor. Error bars should be added to identify whether these performance improvements are significant and how robust to different random seeds is the model compared to the other baselines.
>
> We extended our tables with mean and std of the metrics. Additional for Harbor dataset we added PRAUC as an additional metric since the dataset is heavily imbalanced (1:20 for the positive class)
>
> >Q1. How does the Time-equivariant Contrastive Learning (TC) method handle highly irregular intervals between patient visits?
>
> The SIGF datasets consist of color fundus scans with highly irregular time intervals. The average time interval is 1.67 years with a standard deviation of 1.34. It is important to note that the highest interval is 13 years. This is in contrast with the HARBOR dataset, which consists of monthly acquired scans over 2 years. In Table 1, TC performs better with a larger margin compared to the other equivariance methods. We believe this highlights TC’s robustness to irregular intervals.
>
> >Q5. What are the computational complexities involved in the TC method, especially when applied to large-scale datasets?
>
> Main complexity of TC and the other equivariant contrastive methods for time, is the pair forming scheme. In each epoch, a patient only provides a single randomly formed pair. Ideally, the number of epochs should be sufficiently large that a network is optimized multiple times with a certain pair. In our set-up, we select the time difference between 30-270 days for the HARBOR dataset. It yields 180 possible pairs per patient. We believe 300 epochs are sufficient for the given number of pairs. Obviously, a larger dataset with more time points and wider possible pair selection intervals requires up-scaling the number of epochs.

---

> > ### Comment · Reviewer_m6pm · 2023-11-23
> >
> > Thank you very much for the detailed responses. My overall rating remains.

---

### Official Review · Reviewer_G1yR · 2023-10-31

**Soundness:** 3 good
**Presentation:** 4 excellent
**Contribution:** 3 good
**Rating:** 5
**Confidence:** 3

**Summary:**

Authors propose an approach towards contrastive learning that allows for learning representations across time in images that represent disease progression. From this they propose modifications to contrastive objectives to ensure that the learned representations identify what remains static versus what changes longitudinally in order to help predict disease progression.

**Strengths:**

- This paper has clear clinical significance and as a result the modification to the image-based contrastive objectives is clear.

 - The addition of the regularizing term to prevent collapse is interesting and the effectiveness of it is demonstrated in experimental results

**Weaknesses:**

- It is unclear how the hyperparameters are selected. Specifically, for VICReg, hyperparameters are selected to bring the magnitude of the loss terms in the same range. What does this mean? What happens when you use the hyperparameters from the original VICReg implementation?

 - The authors note that the displacement term needs to be larger than 0 in order to prevent collapse. However, in many medical settings it is possible that there is no change. However, it looks like this model requires a small amount of change. Since the severity of the disease is monotonically non-decreasing, it seems like it is also the case here. If that is true, this method seems very limited in its application.

 - I believe that the introduction could be improved. Certain elements seem to be out of order. This makes it difficult to understand what the specific issue is that you’re trying to address.

**Questions:**

- How were hyperparameters selected? Specifcally, could you provide some intuition for why you selected the hyperparameters you chose?

---

> ### Author Response · Authors · 2023-11-23
> **Comments on Weaknesses**
>
> Thank you for your time and your comments. We addressed your concerns mentioned in the Weaknesses:
>
> > It is unclear how the hyperparameters are selected. Specifically, for VICReg, hyperparameters are selected to bring the magnitude of the loss terms in the same range. What does this mean? What happens when you use the hyperparameters from the original VICReg implementation?
>
> In pretext tasks such as the rotation prediction, it is possible to monitor the pretext task accuracy performance and optimize the hyperparameters accordingly. But the contrastive methods do not rely on a pretext task. In order to avoid biasing for the supervised downstream task, we only monitored the values of the loss terms during pretraining and ensured that they converges in a similar rate.
>
> > The authors note that the displacement term needs to be larger than 0 in order to prevent collapse. However, in many medical settings it is possible that there is no change. However, it looks like this model requires a small amount of change. Since the severity of the disease is monotonically non-decreasing, it seems like it is also the case here. If that is true, this method seems very limited in its application.
>
> It is corrected that the model requires a change, the displacement map still can take very small values. In our setup, we targeted degenerative diseases that affect the retina. In AMD, the patient monitoring begins with a diagnosis of the intermediate stage. And the disease only progresses from that point. Although the application seems limited, AMD is the leading cause of blindness, affecting millions in the world. There is not much clinical insight on the biomarkers that cause the stage conversion. This makes good predictive models necessary for timely delivery of the available drugs and reducing the monitoring load of the hospitals

---

> > ### Comment · Reviewer_mWX7 · 2023-12-03
> > **Response to the comment**
> >
> > We appreciate the authors' response, yet there are several concerns that need addressing:
> > - The authors acknowledge the underperformance of the current network structure. It would be beneficial if they conducted the suggested experiments to develop a more specific network and compare those results.
> > - The authors’ proposed training strategy seems to focus more on cycle prediction, to demonstrate the model's comprehension of 'time interval,' akin to what is seen in cycleGAN with its cycled prediction. This approach doesn't seem to target predicting past images, which might limit its applicability in certain scenarios.
> > - The authors' admission of requiring 11 days for pretraining is a considerable drawback, especially considering the dataset's relatively small size. This duration seems excessive and raises questions about the method's efficiency.
> > - Reviewers m6pm and 9udP have raised valid points regarding the computational complexities and extended duration of pre-training. An 11-day pre-training period is generally seen as impractical and inefficient.
> > - Both Reviewers G1yR and m6pm have noted issues related to the selection of hyperparameters, given the multiple loss items involved. Conducting a comprehensive ablation study to explore this aspect further would be highly beneficial.
> >
> > Thus, I keep my original rating.

---

### Official Review · Reviewer_9udP · 2023-11-01

**Soundness:** 2 fair
**Presentation:** 1 poor
**Contribution:** 2 fair
**Rating:** 3
**Confidence:** 4

**Summary:**

This paper proposes time-equivariant contrastive learning (abbreviated TC by the authors) and applies this method to longitudinal medical image analysis. A key component of TC is an equivariance module. Given two unlabeled scans of the same patient as well as the time difference between them, this equivariance module is trained to predict the later later scan's representation using the earlier scan and the time duration from the earlier scan to the later scan's time. TC outperforms various equivariant contrastive baselines on two longitudinal ophthalmic imaging datasets.

**Strengths:**

- The medical application considered is compelling.
- The proposed method appears to achieve highly competitive accuracy scores compared to the baselines evaluated.
- I found Figure 1 very helpful.

**Weaknesses:**

- The exposition currently is quite muddled. For example, already in Section 1, I find there to quickly be a lot of unnecessary details making it difficult to tease out what exactly the key takeaways are. For Section 1 anyways, I'd suggest reworking the exposition to more clearly get at what the key ideas of the proposed method are, and what limitations of existing work the proposed method aims to address.
- In the first paragraph, the text says that "However, sensitivity to some of these transformations may be crucial for specific downstream tasks, such as color information for flower classification or rotation for traffic sign detection." This is in reference to, if I understand correctly, how data augmentation is used standardly in contrastive learning (e.g., SimCLR, supervised contrastive learning). However, when using such contrastive learning approaches, it is standard to make sure that the random perturbations/transformations used for data augmentation only consist of changes that the model shouldn't care about whereas the ones that we actually want the latent embedding representation to learn should not be used in data augmentation. Am I missing something here?
- How does the proposed method TC relate to steerable equivariant representation learning (Bhardwaj et al 2023)?
- Please update the experimental setup so that error bars could be reported (such as running experimental repeats with different random seeds and reporting mean/std dev of different achieved evaluation scores).
- Overall I would like to see a much more detailed discussion of precisely why the authors think that the proposed method at times outperforms existing 3 equivariant contrastive methods.
- There is a large body of literature on medical image registration. I think some discussion of this literature would be helpful --- do state-of-the-art methods from this literature simply not work well for the problem setup considered here?
- There are many English issues. Please proofread carefully.
- Please use \citet and \citep appropriately.
- Much more cleanly delineating what already exists vs what the innovations of this paper are would be extremely helpful.

**Questions:**

Please see "weaknesses".

---

> ### Author Response · Authors · 2023-11-23
> **Comments on Weaknesses**
>
> We would like to thank you for your constructive review. We improved the paper and answered some of your questions directly.
>
> > In the first paragraph, the text says that "However, sensitivity to some of these transformations may be crucial for specific downstream tasks, such as color information for flower classification or rotation for traffic sign detection." This is in reference to, if I understand correctly, how data augmentation is used standardly in contrastive learning (e.g., SimCLR, supervised contrastive learning). However, when using such contrastive learning approaches, it is standard to make sure that the random perturbations/transformations used for data augmentation only consist of changes that the model shouldn't care about whereas the ones that we actually want the latent embedding representation to learn should not be used in data augmentation. Am I missing something here?
>
> Since contrastive learning is an unsupervised step, it is hard to determine a common set of image perturbations that are not relevant for all possible downstream tasks. To give an example, in  "What should not be contrastive in contrastive learning."  Xiao, Tete, et al. showed that invariance to color perturbations is harmful when classifying flowers, while it is useful in coarse-grained animal classification task. Similarly, we can argue that rotation invariance is harmful for the traffic sign classification since some signs differ only in orientation, but their classes are different. Both tasks can be defined on the ImageNet dataset. But when using contrastive pretraining, there should be a bias for the target downstream task to prevent unintended invariances.  Similar motivation can be found in ESSL, AugSelf, and EquiMod. Since our focus is on progression of the degenerative diseases, we hypothesized that time sensitive representations are useful in predictive downstream tasks.
>
> > How does the proposed method TC relate to steerable equivariant representation learning (Bhardwaj et al 2023)?
>
> Thanks for highlighting another related work. It tries to find a balance between invariance and equivariance within the same representation space. We believe that this approach is hard to optimize, since learning complete invariance is the trial solution for the equivariance. EquiMod solves this problem by enforcing invariance and equivariance in 2 distinct projection spaces after the representations. But it requires calculation of 2 different costly contrastive losses. We argued that the second projection space is unnecessary, such that if the representation space is equivariant, this space can be projected to an invariant space trivially on which we calculate the contrastive task for the invariance. Also we proposed the displacement map regularization term instead of a second contrastive task (similar to EquiMod).
>
> > Please update the experimental setup so that error bars could be reported (such as running experimental repeats with different random seeds and reporting mean/std dev of different achieved evaluation scores).
>
> We extended our tables with mean and std of the metrics. Additional for Harbor dataset we added PRAUC as an additional metric since the dataset is heavily imbalanced (1:20 for the positive class)
>
> > Overall I would like to see a much more detailed discussion of precisely why the authors think that the proposed method at times outperforms existing 3 equivariant contrastive methods.
>
> Firstly, we believe our method enforces equivariance directly by predicting the future representation from a previous one, similar to EquiMod. The transformation  parameter recovery methods, ESSL and SelfAugs, do not guarantee that kind of relation in the representational space. Secondly, we formulated directly on the representation space rather than on a projection space as it is in the EquiMod, thus our representations are directly optimized to be equivariant.
>
> > There is a large body of literature on medical image registration. I think some discussion of this literature would be helpful --- do state-of-the-art methods from this literature simply not work well for the problem setup considered here?
>
> In deep learning, medical image registration can be performed either by learning a linear affine transformation or generating the registered images using generative models. Our main goal is to learn time equivariant representations such that they are useful in the downstream task without generating any image and relying on any labels. We showed that the time equivariance property is useful for the disease progression task, on top of invariance to random perturbations.

---

### Meta-Review · Area_Chair_EiYe · 2023-12-02

**Metareview:**

This submission receives the following ratings: 5, 3, 5, 5, which indicates all the reviewers agree this submission does not meet the publication bar for ICLR.

The manuscript introduces Time-equivariant Contrastive Learning (TC), a novel approach for disease progression analysis in medical imaging. TC leverages equivariant contrastive learning to generate temporally sensitive representations from longitudinal medical images. Key components include an equivariant module for future representation prediction, a regularizing term to prevent representational collapse, and the construction of contrastive pairs across patient visits. The method demonstrated superior performance on two ophthalmic imaging datasets, showcasing its potential in medical applications.

The strengths (Why Not Lower) and weakness (Why Not Higher) are provided below. The reasons in "Why Not Higher" section collectively suggest that the paper, while presenting an interesting idea, may not yet meet the high standards of rigor, clarity, and broad applicability expected for acceptance in ICLR. Further refinement,  expansion of the scope and a more comprehensive analysis could potentially address these concerns in future submissions.

**Justification For Why Not Higher Score:**

- Limited Applicability: The method primarily targets degenerative diseases, potentially restricting broader applications in medical imaging. The assumption of monotonic disease progression may not hold across various conditions.
- Clarity and Presentation: Several reviewers pointed out issues with the paper's exposition, suggesting that it could benefit from clearer delineation of the novel contributions versus existing work.
- Comparative Analysis: A more comprehensive comparative analysis with a broader range of conditions and datasets would strengthen the paper's claims. Additionally, the paper would benefit from a deeper discussion on medical image registration literature and how TC relates to state-of-the-art methods in this domain.
- Technical Details and Experimentation: There were concerns about the selection of hyperparameters, handling of no-change scenarios in medical conditions, and the computational complexity of the method. Addressing these aspects more thoroughly could enhance the paper's robustness.
- Potential Bias and Generalization: The reliance on irreversible disease progression as a model assumption could introduce bias, affecting the method's applicability to a wider range of diseases with variable trajectories.

**Justification For Why Not Lower Score:**

- Interesting Approach: The introduction of time-equivariant contrastive learning in medical imaging is a step forward, particularly for tracking disease progression.
- Clinical Significance: The paper addresses a critical area in medical diagnostics – the ability to predict disease progression, which has immense potential in improving patient care and treatment planning.
- Fair Empirical Results: The method outperforms existing state-of-the-art equivariant contrastive methods on the evaluated datasets, highlighting its effectiveness.
- Technical Depth: The paper presents a well-thought-out methodology, combining aspects of contrastive learning with temporal equivariance, which is technically sound and novel.

---

### Decision · Program_Chairs · 2024-01-16

Reject